# The Molecular Epidemiology of Hepatitis B Virus and Its Resistance-Associated Mutations in the Polymerase Gene in the Americas

**DOI:** 10.3390/microorganisms13081913

**Published:** 2025-08-16

**Authors:** Itzel A. Ruvalcaba, Carlos Daniel Diaz-Palomera, Adrián Alejandro Silva-Ríos, José Francisco Muñoz-Valle, Oliver Viera-Segura

**Affiliations:** 1Licenciatura en Médico Cirujano y Partero, Centro Universitario de Ciencias de la Salud, Universidad de Guadalajara, Guadalajara 44340, Mexico; itzel.ruvalcaba7839@alumnos.udg.mx (I.A.R.);; 2Laboratorio de Investigación en Cáncer e Infecciones, Departamento de Microbiología y Patología, Centro Universitario de Ciencias de la Salud, Universidad de Guadalajara, Guadalajara 44340, Mexico; daniel.diaz@academicos.udg.mx; 3Instituto en Investigación en Ciencias Biomédicas, Centro Universitario de Ciencias de la Salud, Universidad de Guadalajara, Guadalajara 44340, Mexico; biologiamolecular@hotmail.com

**Keywords:** hepatitis B virus, genotypes, RAM, resistance, nucleotide analogs

## Abstract

The hepatitis B virus (HBV) is a DNA virus of major public health concern whose error-prone polymerase has driven the emergence of ten distinct genotypes and a multitude of resistance-associated mutations (RAMs). Herein, we conducted a retrospective observational study analyzing 8152 hepatitis B virus (HBV) sequences from 27 regions across the Americas, retrieved from GenBank, to construct a database and examine associations among HBV genotypes/subtypes, geographic distribution, resistance-associated mutations (RAMs), and resistance to nucleos(t)ide analogs (NAs) used in the treatment of chronic infection. Following phylogenetic analysis, mutations at clinically relevant sites in the reverse transcriptase domain were identified and classified by resistance to NAs. Genotypes A (21.1% A2 and 14.7% A1) and D predominated across the retrieved database, whereas genotypes E, G, H, and I each accounted for fewer than 3% of the sequences. Among the sequences in the database, 10.6% harbored RAMs, with genotypes G, A, and H predominating in this category. The most frequently observed RAM was L180M + M204V/I, which is associated with resistance to LMV, ETV, and TBV, whereas resistance to ADV and TDF remained rare. Genotypes G and A2 were significantly associated with a higher likelihood of harboring multiple RAMs (as evaluated by logistic regression), along with an increased risk of resistance to LMV, ETV, and TBV; the opposite was true for subtype A1. Notably, genotypes H and B5 were associated with an elevated risk of TDF resistance. A comprehensive understanding of RAMs and circulating genotypes in the Americas is essential for identifying high-risk populations and establishing geographically targeted therapeutic strategies.

## 1. Introduction

The World Health Organization (WHO) recognizes hepatitis B as a major public health problem, responsible for approximately 5 million infections in the Americas, with only a 4.4% treatment coverage and a 20,000 annual mortality rate [1]. The hepatitis B virus (HBV) belongs to the *Hepadnaviridae* family and can be transmitted either horizontally or vertically, with perinatal transmission being the predominant route in endemic countries [2]. Following infection, HBV undergoes an incubation period of one to four months, during which approximately 1/3rd of individuals develop symptoms, and 0.5% experience fulminant hepatic failure [3]. After initial acute exposure, the risk of chronicity is 90% in neonates, 30% in children infected at 1–4 years of age, and <5% in those infected in adulthood [1].

HBV is an enveloped icosahedral virus with a partially double-stranded, relaxed circular DNA (rcDNA) genome, approximately 3.2 kb in length, generated through reverse transcription [4]. The structure consists of four overlapping open reading frames (ORFs) from which viral proteins required for replication and pathogenesis are produced: pre-core/core (pre-C/C), DNA polymerase (P), transcriptional co-activator (X), and surface proteins (pre-S/S). Mutations in these regions can result in immune escape (C and S), drug resistance (P), oncogenicity (X and S), vaccine failure (S), and occult infection (S) [5]. Due to its limited proofreading capacity and error-prone polymerase, HBV has a mutation rate of 10^−4^ to 10^−5^ substitutions/site/year, 10-fold higher than that of other DNA viruses, and close to those of RNA viruses, which present mutation rates from 10^−4^ to 10^−6^ [5]. This contributes to extensive viral diversity, with quasispecies, subtypes, and genotypes continually emerging. Currently, 10 HBV genotypes (A–J) and over 40 subtypes have been identified, differentiated by nucleotide divergence rates of >7.5% and 4–7.5%, respectively [6].

Current evidence suggests that the most recent common ancestor of all HBV lineages existed between 20,000 and 12,000 years ago, facilitating the global distribution of early genotypes (A and D) [7]. Genotype A includes eight subtypes, predominantly found in Europe, South Africa, and Brazil. Genotype D comprises twelve subtypes, with a geographic distribution centered in the Mediterranean, Russia, and India [6]. Genotype G has been described as an atypical genotype, whose re-emergence after millennia of low persistence is commonly associated with the human immunodeficiency virus (HIV) pandemic [8]. The modern strain of genotype H likely originated in México, subsequently spreading to Nicaragua in the 1960s and, later, to more distant countries [9]. Genotype F can be divided into six subtypes (F1a–c, F2a–b, and F3–6), all with different areas of concentration in Central and South America. Genotype I arose from a recombination event between genotypes C, G, and A and accounts for only 0.32% of the global burden of chronic infections, mostly centered in East Asia [10]. Genotype E is highly localized in West Africa but is estimated to account for 20% of total chronic infections [11]. Genotype J is a recombinant of genotype C4 (typically in Australian Aborigines) and HBV from gibbons/orangutans, possibly stemming from the island of Borneo during World War II [11,12]. Genotypes B (B1–10) and C (C1–17) are primarily concentrated in Asia [13]; however, genotype B is also found in the Arctic Circle, and genotype C in the Pacific islands [6]. It is important to note that these genotypic differences may lead to variations in immune escape potential, oncogenicity, and responses to treatment, depending on the prevailing selective pressures [14].

The ability of HBV to develop resistance-associated mutations (RAMs) that reduce susceptibility to antiviral drugs used in treatment regimens is well documented. This propensity for resistance development is significantly influenced by HBV’s high substitution rate, which, coupled with an exceptionally high replication rate (peaking at 10^13^ virions per day), makes the daily generation of all possible single-nucleotide mutations and most double mutations mathematically probable [15]. Consequently, a single infected individual can harbor a diverse population of genetic variants known as quasispecies. While RAMs can exist as pre-treatment quasispecies within an individual, they can become dominant under the selective pressure imposed by antiviral therapy. In HBV, these mutations are found in the reverse transcriptase (RT) domain and can be classified as primary, compensatory, putative, and pre-treatment mutations. Within these categories, 42 RAMs have been identified in the HBV polymerase [16].

The primary objectives of antiviral treatment are to prevent progression to cirrhosis and hepatocellular carcinoma. The recommended first-line therapy consists of oral nucleos(t)ide analogs (NAs) with a high genetic barrier and a low rate of resistance associated with long-term use. Depending on their activation mechanism, these can be categorized into nucleotide analogs (adefovir and tenofovir) and nucleoside analogs (lamivudine, telbivudine, and entecavir), which function by causing DNA chain termination. Long-term treatments for chronic HBV infection may become ineffective due to the emergence of viral mutations conferring drug resistance [17,18,19]. Existing evidence indicates that resistance to NAs increases with duration of use, with older antivirals showing higher rates of resistance and cross-resistance after prolonged administration [20,21,22,23,24]. Also, unlike the hepatitis C virus, for which we already have a cure, the cccDNA from HBV can integrate into the host nucleus, resisting antivirals [25]. Genotypes and subtypes may vary in their disease presentation, outcome severity, and resistance patterns to NAs. Molecular epidemiological studies focusing on the burden of RAMs could support the development of public health interventions and guide early therapeutic adjustments in patients at high risk for resistance.

## 2. Methods

### 2.1. Study Design

This retrospective observational study aimed to examine the relationship between the HBV genotype (and their subtypes), geographic origin, specific mutations (S106C, H126Y, N236T, L269I, D134E, A181T/V, A194T, S202C/G/I, V173L, M204V/I, L269I + H126Y, L180M + M204V/I, L180M + M204V/I + V173L, and L180M + M204V/I + I169T), and resistance to five nucleos(t)ide analogs (NAs) approved for the treatment of chronic infection: lamivudine (LMV), adefovir (ADV), entecavir (ETV), telbivudine (TBV), and tenofovir (TDF and TAF) (Figure 1). Emtricitabine (FTC) was not included in this study as it is not yet FDA-approved for HBV treatment, with reports of severe exacerbations upon discontinuation in HIV co-infected patients [26].

The database includes 8152 HBV sequences and corresponding metadata (e.g., genotype, year of isolation, geographic origin, mutations, and resistance) obtained from the public NCBI GenBank database (https://www.ncbi.nlm.nih.gov/genbank/ [accessed on 30 March 2023]). Information on comorbidities or prior antiviral treatment of the patients from whom viral sequencing samples were obtained was extracted from associated publications. Sequences were retrieved between January and March 2023 using the keywords “hepatitis B virus” with the [organism] specification, followed by the Boolean operator “AND” and the corresponding country name ([country]). Clone sequences and those whose chronological or geographical information was incomplete or foreign to the American continent were excluded. Additionally, those sequences with an incomplete or absent reverse transcriptase domain were removed from the database.

### 2.2. Sequence Classification and Phylogenetic Analysis

The sequences were classified into three different groups:Cluster 1: Genotype and subtype identified in GenBank or associated publication (*n* = 3358).Cluster 2: Genotype provided; subtype not reported (*n* = 4071).Cluster 3: Neither genotype nor subtype reported (*n* = 723).

All three groups were further divided by country and region of genomic coverage (complete or partial). The 306 resulting groups were subjected to multiple alignment using the ClustalW v.2.1 program on the Galaxy platform (https://usegalaxy.org/ [accessed on 30 March 2023]). Alignments were manually verified, and in cases requiring correction, MEGA X v.10.2.6 (Pennsylvania, PA, USA) and AliView v.1.28 (Uppsala, Sweden) were used.

Cluster 1 was subjected to confirmatory phylogenetic analyses. Of these, 84 sequences were reclassified according to our findings (Appendix A). Cluster 2 underwent phylogenetic analysis to determine subtypes (Appendix A), while cluster 3 was analyzed to determine both genotype (Appendix A) and subtype (Appendix A). A total of 202 reference sequences were used for the genotyping and subgenotyping (Appendix A), encompassing known HBV genotypes A-J and their subtypes, where applicable. For consistency and due to the controversial classification of genotypes A and B [6,27,28], we employed the extended classification scheme, avoiding the grouping of A3, A4, A5, A7, and B3, B5, B7, B8, and B9. In our analyses, subtypes D3 and D6 were grouped due to the phylogenetic similarities displayed and the lack of clarity in their separation while classifying the sequences [29].

Phylogenetic trees were constructed using the maximum-likelihood (ML) algorithm with the Tamura-Nei model and 1000 bootstrap replicates, implemented in MEGA X. Trees were exported in “.nwk” format and subsequently visualized, analyzed, and recorded using FigTree v.1.4.4. The classification of sequences according to genotype, country, and their alignment to the HBV complete genome reference sequence NC_003977.2 can be found in Appendix A. Out of all the sequences analyzed, three could not be accurately genotyped, and four could not be subgenotyped.

### 2.3. Analysis of Resistance-Associated Mutations

The coding sequence (CDS) of each entry was used for translation. Amino acid translation was performed using the HBV polymerase gene, with EDWGPC (in the 336aa position) as the starter and IHTAEL (in the 719aa position) as the termination marker. These positions cover the RT domain of the polymerase gene. This information was used to build a database containing amino acid data for the analysis of clinically relevant sites in the RT domain, specifically positions 106, 126, 134, 169, 173, 180, 181, 194, 202, 204, 236, and 269. The resistance-associated mutations were classified according to their variants, combinations, and specific functions (classical/primary, compensatory, and putative). Of the mutations at the 12 sites, six were in different combinations with each other, as certain primary mutations often pair up with compensatory ones. Associations between mutations and resistance to specific NAs were determined based on previously established evidence.

### 2.4. Statistical Analysis

The figures in this article were created using Lucidchart website (https://www.lucidchart.com), Excel v.2503, PowerPoint v.2503, and RStudio v.4.3.1 (“Beagle Scouts”). The risk of resistance to NAs and mutations according to the genotypes and subtypes was calculated using univariate logistic regression. Those with an HIV co-infection and treatment prior to this study were also analyzed as risk factors.

Genotype frequencies were organized by country, and distribution patterns were assessed using Hierarchical Cluster Analysis, specifically the Ward method. The resulting dendrogram (Appendix A) was used to structure the international grouping. The analyses were carried out with the IBM SPSS v.27 software.

## 3. Results

### 3.1. Geographical Distribution

A total of 8152 sequences were included, originating from 24 countries, 1 U.S. state (Alaska), and 2 territories (Martinique and Greenland) within the Americas. Although the United States contributed the highest number of sequences (2431), South America was the region with the largest overall contribution, primarily due to Brazil and Argentina. While South and North America contributed a comparable number of sequences, Central America and the Caribbean were underrepresented, with only 748 HBV sequences in the database (Figure 2).

Regarding the global circulation of genotypes, genotype A was found in 19 of the 27 territories studied, and genotype D in 16. Genotype H, present in only six countries, exhibited a markedly higher prevalence in México (65%) and Nicaragua (33%) compared with the other American nations (<1%). Genotype G was also identified in only six countries; in five of them, its prevalence was below 4%, whereas in México, it reached 14%.

### 3.2. Molecular Epidemiology

The 8152 analyzed sequences exhibited a heterogeneous distribution, predominantly comprising genotypes A, C, and D. Genotypes E, G, H, and I were each detected at a frequency below 3% (Figure 3). Overall, the predominant subtypes in the database were A2 (21.1%), A1 (14.7%), C2 (7.8%), D3/D6 (6.9%), and C1 (6.3%).

Furthermore, genotypes were grouped based on their frequency of occurrence and visualized using a hierarchical dendrogram. The analysis revealed a tripartite division of countries according to their epidemiological genotype composition (Figure 4). The orange group was characterized by a high prevalence of genotype F (greater than 60%) and included 11 countries from Central and South America. The blue group was the least represented and comprised countries with a wide diversity of genotypes (e.g., the United States and Canada), regardless of the dominant type. The gray group displayed a predominance of either genotype A or genotype D.

### 3.3. Resistance-Associated Mutations

Sequences containing one or more resistance-associated mutations (RAMs) (*n* = 864) were isolated to construct a figure illustrating the relationship between genotype, specific mutations, RAMs, and resistance to treatments. Among all sequences, 5454 exhibited at least one mutation, and 863 were found to carry a documented resistance-associated mutation. The frequency distribution of genotypes, RAMs, and associated resistance to NAs was visualized using a coded Sankey plot (Figure 5). In contrast with genotype A, in which 93% of sequences harbored mutations, 59% of genotype F sequences lacked clinically relevant mutations. The proportion of sequences with RAMs within each genotype generally ranged from 2% to 7%, with notable exceptions: genotype G (31.0%), genotype A (17.6%), and genotype H (12.9%).

Notably, the L180M + M204V/I combination was present in 467 sequences (54.1%) of those with RAMs and was associated with resistance to ETV, LMV, and TBV. Given that the majority of resistance-associated sequences exhibited resistance to multiple drugs (94.5%), resistance to LMV was observed in 840 sequences (97.3%), TBV in 816 (94.6%), ETV in 217 (25.1%), ADV in 29 (3.4%), and tenofovir in 8 (0.1%). Resistance to ETV was consistently accompanied by resistance to both LMV and TBV, while resistance to TBV was always accompanied by resistance to LMV.

The relationship between subgenotypes and resistance-associated mutations (RAMs) was analyzed (Table 1), revealing that 30 subgenotypes were associated with increased or decreased odds of one or more RAMs. Half of these subgenotypes were associated with increased odds for the L269I mutation, including B2, B3, B5, B6, C1, D1, D2, D3/D6, D4, F2a, F2b, F3, F4, F5, and H. Subgenotype A2 was associated with increased odds for five mutations: H126Y (OR = 12.22; 95% CI: 10.78–13.85), V173L (OR = 5.97; 95% CI: 2.89–12.32), L269I + H126Y (OR = 4.35; 95% CI: 3.84–4.93), L180M + M204V/I (OR = 4.77; 95% CI: 3.94–5.77), and L180M + M204V/I + V173L (OR = 15.80; 95% CI: 11.20–22.28). Similarly, genotype G was associated with increased odds for five mutations: H126Y (OR = 1.81; 95% CI: 1.27–2.59), A181T/V (OR = 9.85; 95% CI: 3.36–28.85), M204V/I (OR = 9.56; 95% CI: 5.91–15.49), L269I + H126Y (OR = 11.94; 95% CI: 8.36–17.07), and L180M + M204V/I + V173L (OR = 2.33; 95% CI: 1.13–4.81). In contrast, subgenotype A1 was associated with significantly decreased odds against five mutations: S106C (OR = 0.25; 95% CI: 0.09–0.67), M204V/I (OR = 0.36; 95% CI: 0.19–0.68), L269I (OR = 0.18; 95% CI: 0.14–0.22), L180M + M204V/I (OR = 0.36; 95% CI: 0.19–0.68), and L180M + M204V/I + V173L (OR = 0.26; 95% CI: 0.13–0.51).

The relationship between HBV subgenotypes and increased odds of resistance to specific nucleos(t)ide analogs (NAs) was analyzed (Table 2). Most significant associations indicated decreased odds rather than increased odds. Subgenotype A2 was identified as being associated with significantly increased odds for resistance to entecavir (ETV) (OR = 12.04; 95% CI: 8.82–16.43), lamivudine (LMV) (OR = 5.45; 95% CI: 4.70–6.32), and telbivudine (TBV) (OR = 5.12; 95% CI: 4.41–5.95). In contrast, subgenotype A1 was consistently associated with decreased odds for resistance to these same drugs: ETV (OR = 0.30; 95% CI: 0.17–0.56), LMV (OR = 0.46; 95% CI: 0.36–0.60), and TBV (OR = 0.48; 95% CI: 0.37–0.62). Notably, genotype H (OR = 13.87; 95% CI: 1.69–114.03) and subgenotype B5 (OR = 87.24; 95% CI: 16.94–449.14) were associated with strongly increased odds for resistance to Tenofovir. Additionally, genotype G was associated with increased odds of resistance to four out of the five NAs evaluated: adefovir (ADV), ETV, LMV, and TBV.

### 3.4. Resistance in Prior Treatments and Co-Infection

The relationship between prior or ongoing treatment with lamivudine (LMV) and tenofovir (TDF) and subsequent resistance to specific nucleos(t)ide analogs (NAs) was analyzed (Figure 6). A total of 83 patient sequences were associated with previous or current use of LMV, and 17 with previous or current use of TDF. Prior treatment with LMV was associated with increased odds for the following mutations: L180M + M204V/I + V173L (OR = 7.32; 95% CI: 3.98–13.45), L180M + M204V/I (OR = 20.98; 95% CI: 13.34–33.01), and H126Y (OR = 6.70; 95% CI: 4.27–10.52). Similarly, prior treatment with TDF was associated with increased odds for L180M+M204V/I (OR = 14.17; 95% CI: 5.20–38.57) and H126Y (OR = 4.16; 95% CI: 1.60–10.81).

Additionally, the relationship between HIV co-infection (present in 428 sequences) and resistance to nucleos(t)ide analogs (NAs) was analyzed (Figure 7), but no statistically significant associations were found (*p* ≥ 0.05).

## 4. Discussion

The molecular epidemiological findings in this study reinforce previous reports on the geographical distribution of HBV genotypes while offering a more comprehensive view of the epidemiological landscape in the Americas. As expected, and consistent with global patterns [6], genotypes A and D were the most widely distributed, present in over half of the countries analyzed. Although subtype A1 has been associated with South America and A2 with North America, due to differing historical routes of spread (the transatlantic slave trade vs. European migration) and modes of transmission (perinatal vs. parenteral/sexual) [6,30]. Our data revealed an inverse pattern in countries such as Argentina, Venezuela, Cuba, and Canada. Similarly, while subtype C1 predominates in Southeast Asia and C2 in East Asia [31], elevated frequencies observed in countries with significant Asian immigration (e.g., the USA, Canada, and Panama) may reflect increasing globalization and shifts in the genetic landscape [32,33].

The regional genotypic profiles also highlighted the influence of endemic strains. As anticipated, genotype F—particularly subtype F1b—was more prevalent in South America, while genotype H showed markedly higher frequencies in México and Nicaragua, likely due to its evolutionary origin and adaptation to local populations [9,34]. These differences between the south of the continent and México can be dated to around 8140 years ago, where genotypes F and H suffered a speciation event. It also predicted that these genotypes started their diversification around 4551 and 2070 years ago, information that is supported by the fact that genotype H was found in native populations [34]. Although historically concentrated in México and considered endemic [8], genotype G is now believed to represent the re-emergence of an ancient Eurasian strain [7]. Its low prevalence in most countries may be underestimated due to its frequent occurrence as a co-infection with other genotypes, complicating diagnosis. Despite its modest representation, genotype G warrants attention due to its high mutation frequency and established link with HIV co-infection, both of which increase the likelihood of RAM development [14].

As expected, genotype J was absent from the dataset, as it has only been documented in a single Japanese individual [12]. Nevertheless, genotypes not typically endemic to the continent were observed. For example, subtype I2—usually restricted to Southeast Asia [35,36]—was detected in a Canadian sample, likely originating from a Vietnamese immigrant [37]. Genotype E, though not dominant in any country, was detected in moderate frequencies in areas with African ancestry (e.g., Haiti and Martinique) or migration (e.g., USA and Canada), consistent with its origin in West Africa [38].

Among the 11 codons analyzed in the reverse transcriptase (RT) domain, several mutation combinations demonstrated that while single mutations are more common, they are less effective in conferring resistance. Notably, clinically relevant mutations can arise spontaneously, independent of antiviral exposure [17], and phylogenetic analyses have traced M204V and L180M mutations back to the mid-20th century [39]. Their early emergence likely accounts for their high prevalence in this study (53.5% and 44.3% of RAM-positive sequences, respectively). The combined M204I/V and L180M mutations were found in 2.7% of sequences, aligning with global estimates [16,39,40]. One sequence harboring the A181T/V mutation—associated with ADV resistance [14,16,41]—was collected 15 years before the approval of the first HBV treatment [42], highlighting how high mutation rates can yield clinically relevant variants even in the absence of selective drug pressure.

Of all sequences with RAMs, 94.5% conferred resistance to more than one antiviral agent, typically LMV, TBV, and ETV. These findings confirm that the HBV mutation pool continues to evolve in response to the selective pressures exerted by successive antiviral therapies. The risk of resistance increases with prolonged NA exposure [20,23,24], and cross-resistance—particularly following LMV treatment—is well documented [21,22]. In our study, all sequences resistant to ETV also displayed resistance to LMV and TBV, while all TBV-resistant sequences were also LMV-resistant.

Consistent with global prevalence estimates, resistance to tenofovir was detected in only 0.1% of sequences. However, subtypes B5 and H were associated with the A194T mutation, which confers resistance to TDF. Although both had the highest rates of TDF- and ETV-resistant sequences, the endemic nature and genetic adaptation of genotype H to the Mexican population [34] complicate the interpretation of its association analysis. While LMV and TBV showed the highest number of significant associations across subtypes (13 and 14, respectively), this could be attributed to their high representation in the database.

Despite A194T’s known association with TDF resistance, this drug typically requires specific mutational patterns to lose efficacy [39]. Two previously reported quadruple-mutation combinations may contribute to resistance development: CYEI (S106C + H126Y + D134E + L269I) and MLVV (L180M + T184L + A200V + M204V) [43]. Although L269I alone does not confer resistance, it was associated with an increased odds ratio in 15 subtypes. Moreover, three subtypes (A1, A2, and G) were linked to its combination with S106C and H126Y (CYI). The L180M + M204V/I mutation, associated with resistance to LMV, TBV, and ETV, was present in 54.1% of resistant sequences. These combinations highlight the need for further research on mutation synergy and its impact on resistance.

Genotype G emerged as the genotype with the highest association with five mutations (H126Y, A181T/V, M204V/I, L269I + H126Y, and L180M + M204V/I + V173L) and resistance to four of the five NAs analyzed (ADV, ETV, LMV, and TBV). This aligns with the prior literature linking genotype G to an increased number of RAMs and NA resistance [14,44]. Given its strong association with HIV risk groups (e.g., intravenous drug users and men who have sex with men) [45,46,47,48,49], extensive NA use in these populations may exert pre-infection selective pressure, promoting the emergence of RAMs with cross-resistance to HIV and HBV therapies. Conversely, genotype E had the lowest RAM prevalence, consistent with its regional distribution in areas with minimal antiviral coverage (0.2%) and limited public health funding [1], which reduces the selective pressure for RAM development.

Intra-genotypic differences were also evident. Subtype A2 was associated with resistance to LMV, ETV, and TBV, while subtype A1 was protective against resistance to the same drugs. This contrast, previously reported in the literature [14], may reflect their differing geographic and historical contexts [6,30]. Subtype A1’s distribution in countries with limited treatment access likely contributes to reduced selective pressure, while A2’s higher replication capacity [30] may increase its likelihood of developing RAMs.

Our findings also support established evidence that prior exposure to antiviral therapy increases the risk of resistance [20]. Sequences from patients previously treated with LMV and TDF showed increased odds of carrying mutations (L180M, M204V/I, H126Y, and/or V173L) associated with resistance to LMV, TBV, and ETV.

Although the number of sequences obtained from each region (North, Central, and South America) was proportional to its population [50], some countries were disproportionately represented (e.g., Cuba in the Caribbean and Argentina in South America), while others were underrepresented (e.g., Guatemala in Central America and Ecuador in South America). This may result in a skewed perception of regional genotypic distribution. Countries like the USA, with abundant sequencing data, provide a clearer epidemiological picture, while nations like Guatemala or Ecuador require more representative sampling. These disparities highlight the need for enhanced molecular surveillance in underrepresented areas.

A key distinction between this study and broader global analyses [39] is the focus on genotypes more prevalent in the Americas (e.g., H, G, and F), which are often underrepresented in global datasets due to the dominance of European (A and D) and Asian (B and C) genotypes. This information supports the adaptation of treatment strategies based on geographic profiles, even in settings where routine genotyping is unavailable. In highly heterogeneous countries such as Canada, the United States, and Suriname, national-level surveillance may be insufficient, and regional assessments may be necessary. In resource-limited settings, the local molecular epidemiology may guide drug prioritization and resistance risk management. Although the 2024 WHO guidelines recommend TDF as the first-line agent due to its high resistance barrier, often in combination with LMV or FTC when monotherapy is not feasible [2], the availability of these NAs remains limited in many low-resource countries. Furthermore, an inherent limitation of our study is the lack of complete clinical meta-information associated with publicly available HBV sequences. Specifically, we were unable to determine whether the sequences were obtained before or after antiviral therapy. This limitation prevents an analysis that directly compares the frequency of pre- and post-treatment mutations, a critical point for discerning between mutations emerging due to drug selection pressure and pre-existing polymorphisms.

While progress has been made in managing HBV across the region, emerging therapies in early development stages [51], further efforts are needed to address drug resistance effectively. Understanding the molecular epidemiology of HBV across the continent is crucial for estimating clinical implications, particularly in light of established genotype–resistance associations. A comprehensive genotypic analysis of resistance mutations can inform public health policies and improve treatment strategies. Such profiling would enable healthcare professionals to identify patient groups at higher risk for resistance and tailor therapy accordingly.

## Figures and Tables

**Figure 1 microorganisms-13-01913-f001:**
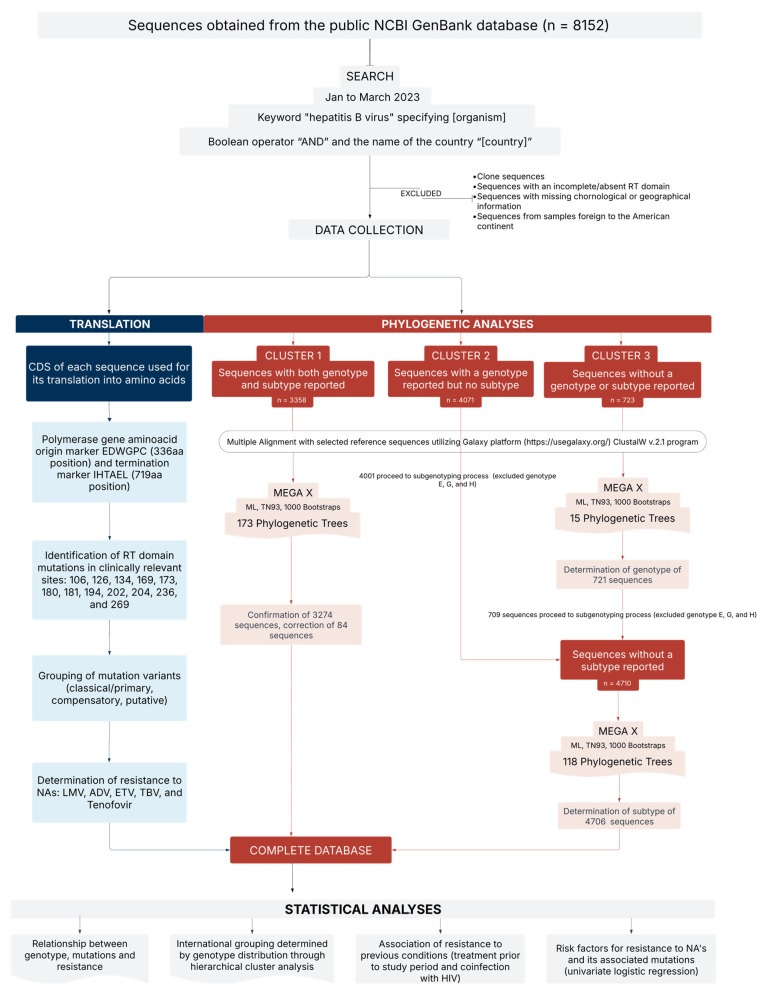
This study analyzed 8152 HBV sequences from the GenBank database to investigate associations among genotypes, mutations, and resistance to the main nucleos(t)ide analogs (NAs) used in the treatment of chronic HBV infection. The sequences were classified into three clusters based on genotype information availability, followed by phylogenetic analyses to determine genotypes and subtypes. Statistical analyses were performed to evaluate resistance-associated risk factors and genotype distribution across countries. ADV, adefovir; CDS, coding region; ETV, entecavir; LMV, lamivudine; ML, maximum likelihood; NAs, nucleo(s)tide analogs; RT, reverse transcriptase; TBV, telbivudine; TN93, Tamura-Nei 1993.

**Figure 2 microorganisms-13-01913-f002:**
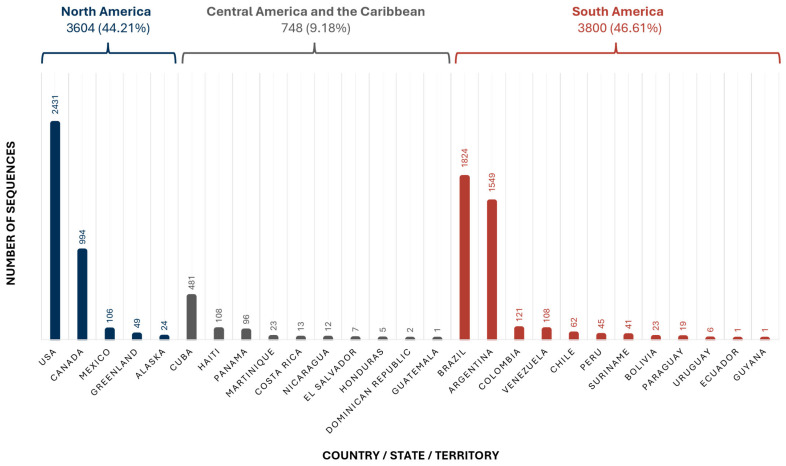
The number of sequences included in the database, classified by region and by country, state, or territory. Alaska was presented separately from the rest of the United States due to the inherent differences in the genetic composition of its population.

**Figure 3 microorganisms-13-01913-f003:**
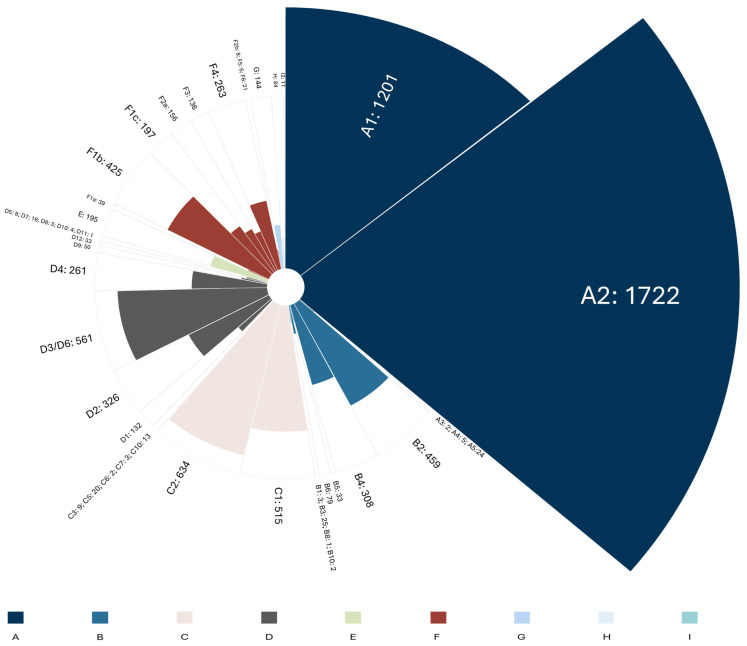
The number of sequences corresponding to each genotype and subtype included in the database. A total of 45 genotypes/subtypes were reported across the Americas. The most frequent subtypes were A2 (*n* = 1722), A1 (*n* = 1201), C2 (*n* = 634), D3/D6 (*n* = 561), C1 (*n* = 515), B2 (*n* = 459), and F1b (*n* = 425). Two sequences identified as natural recombinants (A/G and A/D) were excluded from the figure. Genotype J was not detected, as expected.

**Figure 4 microorganisms-13-01913-f004:**
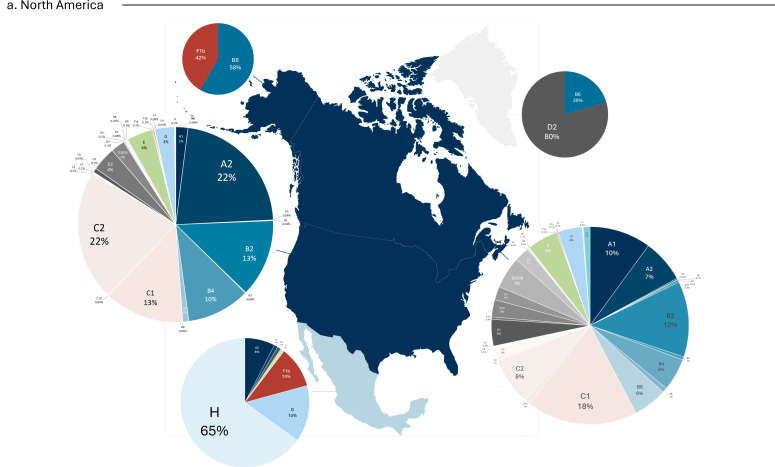
Geographic distribution of HBV subtypes across the Americas, classified by country of origin and region: (**a**) North America, (**b**) Central America and the Caribbean, and (**c**) South America. The map displays the percentage frequencies of subtypes (represented as pie charts) in each of the 27 areas included in the database. Countries marked in blue exhibited a heterogeneous genotype distribution, with varying proportions of genotypes A. Countries and territories in gray showed a prominent presence of either genotype A or D. Countries highlighted in orange had a predominance of genotype F.

**Figure 5 microorganisms-13-01913-f005:**
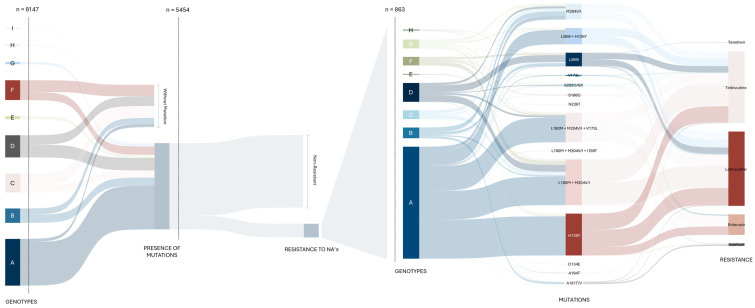
A Sankey plot illustrating the relationship between genotype, mutations, and resistance to nucleos(t)ide analogs (NAs). The figure displays the frequency of genotypes reported in the total dataset (*n* = 8147), the subset with mutations (*n* = 5454), and, among those, the sequences harboring resistance-associated mutations (RAMs) (*n* = 863). For the 863 sequences with RAMs, the distribution of genotypes (A–H), specific RAMs, and resistance to nucleotide analogs (tenofovir and adefovir) and nucleoside analogs (telbivudine, lamivudine, and entecavir) were evaluated. Genotypes G and E, along with genotype A, exhibited the highest proportion of sequences with mutations (100% and 99%, respectively). However, only 16% of sequences with mutations contained RAMs. Genotype A accounted for 60% of all treatment-resistant sequences, representing 92% of all sequences with the H126Y mutation and 63% of those with the L180M + M204V/I mutation. The L180M + M204V/I and H126Y mutations together accounted for more than half of the observed RAMs, while resistance to TBV and LMV was the most frequently reported. Sequences with undetermined genotypes (NA) and recombinants were excluded from the figure.

**Figure 6 microorganisms-13-01913-f006:**
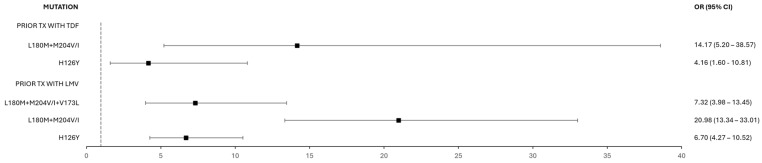
Odds ratios and 95% confidence intervals for the association between prior treatment with lamivudine (LMV) or tenofovir (TDF) and the subsequent presence of resistance-associated mutations (L180M, M204V/I, H126Y, and/or V173L).

**Figure 7 microorganisms-13-01913-f007:**
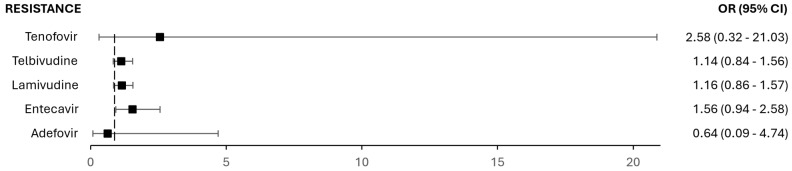
Odds ratios and 95% confidence intervals for resistance to nucleos(t)ide analogs (NAs) in patients with HIV co-infection.

**Table 1 microorganisms-13-01913-t001:** Odds ratios and 95% confidence intervals for significant associations between HBV subtypes and specific resistance-associated mutations (RAMs). Subgenotypes highlighted in pink were identified with increased odds of a mutation, while those in green showed decreased odds. Subgenotypes A2 and genotype G were both associated with significantly increased odds for five mutations, while subgenotypes C1 and H were associated with four. Conversely, subgenotypes D3/D6 and C2 had decreased odds for four mutations each, and A1 had high odds for five. No significant associations were found between any subgenotype and the mutation L180M + M204V/I + I169T; this mutation was, therefore, excluded from the table. NS, not significant (*p* ≥ 0.05); -, no sequences with the specific combination of subgenotype and mutation.

Subtype	S106C	H126Y	D134E	V173L	A181T/V	A194T	S202C/G/I	M204V/I	N236T	L269I	L269I + H126Y	L180M + M204V/I	L180M + M204V/I + V173L
**A1**	0.25 (0.09–0.67)	3.55 (3.11–4.04)	NS	NS	NS	NS	NS	0.36 (0.19–0.68)	-	0.18 (0.14–0.22)	4.64 (4.05–5.31)	0.36 (0.19–0.68)	0.26 (0.13–0.51)
**A2**	-	12.22 (10.78–13.85)	NS	5.97 (2.89–12.32)	-	-	NS	NS	NS	0.02 (0.01–0.03)	4.35 (3.84–4.93)	4.77 (3.94–5.77)	15.80 (11.20–22.28)
**A5**	-	NS	-	-	-	-	-	-	-	-	117.24 (15.82–868.9)	-	-
**B2**	-	-	-	-	NS	NS	-	NS	NS	6.37 (5.20–7.79)	-	0.21 (0.09–0.47)	-
**B3**	-	-	-	-	-	-	-	-	-	8.77 (3.5–22.0)	-	-	-
**B4**	-	0.01 (0.002–0.09)	NS	-	18.12 (8.34–39.39)	-	NS	7.67 (5.21–11.31)	-	NS	-	0.26 (0.11–0.64)	-
**B5**	71.34 (34.6–147.07)	-	-	-	-	87.24 (16.94–449.13)	-	NS	-	6.38 (3.03–13.42)	-	-	-
**B6**	-	-	-	-	NS	-	-	-	-	14. 36 (7.91–26.08)	-	-	-
**B10**	-	-	-	-	-	-	-	-	-	-	-	16.49 (1.03–264.04)	-
**C1**	2.99 (1.74–5.15)	0.01 (0.004–0.06)	5.18 (2.86–9.37)	NS	NS	NS	-	2.03 (1.26–3.27)	-	3.37 (2.82–4.04)	-	0.28 (0.014–0.54)	-
**C2**	NS	0.05 (0.02–0.09)	2.15 (1.05–4.39)	NS	-	-	-	NS	-	0.27 (0.21–0.36)	0.01 (0.004–0.06)	NS	0.06 (0.008–0.41)
**C5**	-	NS	15.74 (3.57–69.42)	-	-	-	-	NS	-	NS	-	-	-
**D1**	-	-	NS	-	4.92 (1.15–20.99)	-	-	NS	30.60 (2.76–339.6)	1.58 (1.11–2.27)	-	NS	-
**D2**	-	-	-	NS	-	-	NS	NS	-	4.90 (3.89–6.16)	-	2.81 (2.02–3.91)	NS
**D3/D6**	NS	0.02 (0.006–0.06)	NS	NS	-	-	3.89 (1.28–11.85)	NS	-	4.34 (3.64–5.18)	0.02 (0.004–0.07)	0.34 (0.19–0.61)	0.13 (0.03–0.53)
**D4**	-	0.02 (0.002–0.11)	-	-	-	NS	-	-	-	3.28 (2.56–4.2)	-	-	-
**D9**	-	-	-	NS	NS	-	-	NS	-	NS	-	NS	4.41 (1.73–11.24)
**D12**	-	-	-	-	-	-	14.89 (1.92–115.28)	-	-	0.18 (0.04–0.74)	-	3.69 (1.52–8.99)	-
**E**	-	2.86 (2.14–3.82)	-	-	-	-	-	NS	-	0.01 (0.002–0.1)	6.97 (5.22–9.31)	0.25 (0.08–0.79)	-
**F1b**	7.03 (4.43–11.14)	-	-	-	NS	-	NS	-	-	NS	0.01 (0.002–0.08)	NS	NS
**F1c**	3.74 (1.79–7.82)	-	-	-	-	-	-	-	-	0.19 (0.11–0.33)	-	-	-
**F2a**	NS	0.03 (0.004–0.18)	-	-	-	-	-	-	-	3.66 (2.66–5.05)	-	NS	-
**F2b**	-	-	-	-	-	-	-	-	-	8.26 (1.67–40.98)	-	5.51 (1.11–27.35)	-
**F3**	7.23 (3.67–14.25)	-	-	NS	-	-	-	NS	-	5.38 (3.77–7.69)	-	NS	-
**F4**	4.39 (2.37–8.14)	-	-	-	-	-	NS	-	-	1.90 (1.48–2.44)	-	0.19 (0.06–0.58)	NS
**F5**	-	-	-	-	-	-	-	-	-	5.50 (1.007–30.07)	-	-	-
**F6**	-	-	-	-	-	-	23.87 (3.03–187.99)	-	-	NS	-	NS	-
**G**	-	1.81 (1.27–2.59)	-	-	9.85 (3.36–28.85)	-	-	9.56 (5.91–15.49)	-	-	11.94 (8.36–17.07)	NS	2.33 (1.13–4.81)
**H**	NS	-	-	6.76 (1.59–28.8)	NS	13.87 (1.69–114.03)	12.27 (2.78–54.24)	NS	-	6.27 (3.94–9.99)	0.06 (0.008–0.43)	NS	NS
**I**	-	NS	-	-	-	-	-	-	-	-	50.48 (6.46–394.69)	-	-

**Table 2 microorganisms-13-01913-t002:** Odds ratios and 95% confidence intervals for significant associations between HBV subgenotypes and resistance to specific nucleos(t)ide analogs (NAs). Subgenotypes highlighted in pink were identified as risk factors for resistance, while those in green were classified as protective factors against resistance to the corresponding NA. Lamivudine and telbivudine showed the highest number of significant associations, with 13 and 14 subgenotypes, respectively. In contrast, resistance to tenofovir was significantly associated only with subgenotypes B5 and H. NS, not significant (*p* ≥ 0.05); -, no sequences with the specific combination of subgenotype and mutation.

Subtype	Adefovir	Entecavir	Lamivudine	Telbivudine	Tenofovir
**A1**	NS	0.30 (0.17–0.56)	0.46 (0.36–0.60)	0.48 (0.37–0.62)	NS
**A2**	0.13 (0.02–0.98)	12.04 (8.82–16.43)	5.45 (4.70–6.32)	5.12 (4.41–5.95)	-
**B2**	NS	-	0.18 (0.10–0.35)	0.19 (0.10–0.36)	NS
**B4**	16.10 (7.54–34.40)	0.12 (0.02–0.82)	NS	1.40 (1–1.96)	-
**B5**	-	-	NS	NS	87.24 (16.94–449.14)
**C1**	NS	-	0.50 (0.34–0.74)	0.52 (0.36–0.76)	NS
**C2**	-	0.05 (0.007–0.38)	0.52 (0.37–0.73)	0.54 (0.38–0.75)	-
**D1**	4.55 (1.07–19.35)	-	NS	NS	-
**D2**	-	NS	1.86 (1.38–2.51)	1.93 (1.43–2.59)	-
**D3/D6**	-	0.38 (0.17–0.86)	0.35 (0.23–0.54)	0.37 (0.24–0.56)	-
**D9**	NS	4.14 (1.63–10.52)	NS	NS	-
**E**	-	-	0.18 (0.07–0.48)	0.18 (0.07–0.50)	-
**F1b**	NS	NS	0.55 (0.37–0.83)	0.57 (0.38–0.86)	-
**F2a**	-	-	0.34 (0.15–0.78)	0.36 (0.16–0.81)	-
**F3**	-	-	0.40 (0.18–0.90)	0.41 (0.18–0.93)	-
**F4**	-	NS	0.13 (0.05–0.35)	0.14 (0.05–0.36)	-
**G**	9.06 (3.11–26.37)	2.18 (1.05–4.50)	3.56 (2.46–5.14)	3.68 (2.54–5.32)	-
**H**	NS	NS	NS	NS	13.87 (1.69–114.03)

## Data Availability

The original contributions presented in this study are included in the article/Appendix A. Further inquiries can be directed to the corresponding author.

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
