# Peer review of "The Molecular Epidemiology of Hepatitis B Virus and Its Resistance-Associated Mutations in the Polymerase Gene in the Americas"

_microorganisms, 2025, doi:10.3390/microorganisms13081913_

Round 1

Reviewer 1 Report

Comments and Suggestions for Authors

This is an interesting, well documented, and well written analysis of of HBV genotypes and drug resistance mutations in America. My critique is relatively minor

HBV has a mutation rate  similar to that of RNA viruses – while HBV has more than 10 fold higher mutation rate than other DNA viruses  it does not reach the level of diversity seen on most RNA viruses. This is due to the overlapping ORFs. Thus, HBV is somewhere between other DNA and RNA viruses. Please correct

It is unclear what sequences were included in the analysis. Whole length? Or any as long as they contained reverse transcriptase domain? Please provide exact length (nt positions). A follow up question is what was the genotyping based on? Some regions like core/precure and polymerase may be less informative than S, pre-S/S  due to overlapping constraints.

Alignments  were manually verified, and in cases requiring correction, MEGA X v.10.2.6 and AliView v.1.28 were used. What corrections were necessary?

When writing that a particular type/subtype was identified as a significant risk factor for drug resistance (in many places throughout the manuscript) – provide evidence for causality or stick to the term association

Author Response

This is an interesting, well documented, and well written analysis of of HBV genotypes and drug resistance mutations in America. My critique is relatively minor

Query 1: HBV has a mutation rate  similar to that of RNA viruses – while HBV has more than 10 fold higher mutation rate than other DNA viruses  it does not reach the level of diversity seen on most RNA viruses. This is due to the overlapping ORFs. Thus, HBV is somewhere between other DNA and RNA viruses. Please correct

Answer 1: This piece of information was updated in the current manuscript version.

Query 2: It is unclear what sequences were included in the analysis. Whole length? Or any as long as they contained reverse transcriptase domain? Please provide exact length (nt positions). A follow up question is what was the genotyping based on? Some regions like core/precure and polymerase may be less informative than S, pre-S/S  due to overlapping constraints.

Answer 2: The authors agree that this information is unclear in the manuscript; to address this, a supplementary Table 2 was included that contains information regarding the size of the selected amplicon and the region between which the problem sequence was aligned to the reference sequence. It is also important to mention that, given that the primary focus of our study was to find the distribution of the mutations of interest reported in the viral reverse transcriptase, we downloaded from GenBank those sequences that contain the studied region, which correspond to the Pre-s/S-gene and overlapping reverse transcriptase.

Query 3: Alignments  were manually verified, and in cases requiring correction, MEGA X v.10.2.6 and AliView v.1.28 were used. What corrections were necessary?

Answer 3: Corrections were minor and case-specific. Most common corrections involve adjusting the position of gaps and misaligned regions to ensure that homologous sites are correctly aligned, trimming or adjusting the ends of sequences, and manually reviewing.

Query4: When writing that a particular type/subtype was identified as a significant risk factor for drug resistance (in many places throughout the manuscript) – provide evidence for causality or stick to the term association

Answer 4: The authors agree with the reviewer on the correctness of the word choice, so the manuscript was updated taking the comment into account.

The authors are deeply grateful for the review of the manuscript.

Reviewer 2 Report

Comments and Suggestions for Authors

The manuscript by Ruvalcaba et al. report a large re-analysis of published HBV sequences covering most countries of the Americas. Data was scrutinised in terms of known mutations related to antiviral resistance.

The introduction should mention that contrary to antiviral HCV drugs, HBV drugs whether nucleotide or nucleoside analogs contain viral replication but do not cure the infection because of the integration of cccDNA so far unaffected by the antiviral drugs currently available.

The authors should also indicate that HBV genotypes F and H appear indigenous to the Americas having been introduced nearly 20,000 years ago for F and H having branched out of F approximately 2000 years ago (Jose-Abrego et al Frontiers in Microbiology 2023). This contrasts with genotypes A, B, C and D that were imported by immigration mostly from Europe and East Asia. Genotype E being posterior to the end of the slave trade remains rare in the Americas.

M&M section 2.2. The authors should indicate whether their computing of genotypes and subtypes was based on full genome or pre-S/S sequences and if a mixture, how many of each.

Sample selection and available information should allow authors to identify sequences obtain before and after antiviral treatment. At least for Lamivudin and tenofovir, the frequency of specific mutations before and after therapy should be available and results presented. Data from other drugs pre- and post-therapy could be compared if available.

In section 3.2 and figure 1, the predominance of genotype A2 is related to the large number of samples tested in the USA and Canada which present a massive bias. It would be more representative to calculate genotype dominance by country  (as in figure 4) or regions such as North America, central America including Mexico, Caribbean islands, Brazil and Argentina and rest of South America. Such dominance could be factored by population size.

Author Response

The manuscript by Ruvalcaba et al. report a large re-analysis of published HBV sequences covering most countries of the Americas. Data was scrutinised in terms of known mutations related to antiviral resistance.

Query1: The introduction should mention that contrary to antiviral HCV drugs, HBV drugs whether nucleotide or nucleoside analogs contain viral replication but do not cure the infection because of the integration of cccDNA so far unaffected by the antiviral drugs currently available.

Answer 1: This piece of information was updated in the revised version of the manuscript.

Query 2: The authors should also indicate that HBV genotypes F and H appear indigenous to the Americas having been introduced nearly 20,000 years ago for F and H having branched out of F approximately 2000 years ago (Jose-Abrego et al Frontiers in Microbiology 2023). This contrasts with genotypes A, B, C and D that were imported by immigration mostly from Europe and East Asia. Genotype E being posterior to the end of the slave trade remains rare in the Americas.

Answer 2: In accordance to the reviewer, this piece of suggested information contributes to the discussion of the manuscript results (lines 429-443).

Query3: M&M section 2.2. The authors should indicate whether their computing of genotypes and subtypes was based on full genome or pre-S/S sequences and if a mixture, how many of each.

Answer 3: The authors agree that this information is unclear in the manuscript; to address this, a supplementary Table 2 was included that contains information regarding the size of the selected amplicon and the region between which the problem sequence was aligned to the reference sequence.

Query 4: Sample selection and available information should allow authors to identify sequences obtain before and after antiviral treatment. At least for Lamivudin and tenofovir, the frequency of specific mutations before and after therapy should be available and results presented. Data from other drugs pre- and post-therapy could be compared if available.

Answer 4: We sincerely appreciate your valuable comments on our manuscript. We agree that the distinction between sequences obtained before and after antiviral treatment is crucial for a more accurate interpretation of the results. Unfortunately, as is common with data from public databases such as GenBank, meta-information on the treatment history of each sample is not consistently available. Our analysis was based on a massive compilation of public sequences, most of which did not include this information. Due to this inherent limitation of the database, it was impossible for us to differentiate between pre- and post-treatment sequences, since most manuscript authors do not report this information. We recognize that this is an important limitation of the study. We have included a new section in the manuscript (or expanded the Limitations section) to address this point, as detailed below. We believe that the analysis, despite this limitation, remains valuable for identifying associations between subgenotypes and mutations, providing a basis for future studies that may include complete clinical data.

Query 5: In section 3.2 and figure 1, the predominance of genotype A2 is related to the large number of samples tested in the USA and Canada which present a massive bias. It would be more representative to calculate genotype dominance by country  (as in figure 4) or regions such as North America, central America including Mexico, Caribbean islands, Brazil and Argentina and rest of South America. Such dominance could be factored by population size.

Answer 5: The reviewer's observation regarding Figure 3 is relevant, given that some countries have a higher proportion of sequences reported in the NCBI database; this biased perception that the data may cause is addressed as a limitation in the discussion section of the manuscript (lines 509-517). Considering the bias that could result from analyzing the data from this point of view, the authors included Figure 4, as the reviewer correctly points out, which offers a more representative view by country of the distribution of each genotype, considering the number of sequences obtained from them. This regionalization is also addressed in the discussion section of the manuscript (lines 418-451; 479-486). These percentages were obtained based on the number of sequences from each country reported in the NCBI. Likewise, the document was updated to avoid these errors of interpretation related to the continental representation of a specific genotype, as this varies by country.

The authors are deeply grateful for the review of the manuscript.

Round 2

Reviewer 2 Report

Comments and Suggestions for Authors

the authors have adequately answered the comments from this reviewer and improved their manuscript by doing so.